# Chronic Use of Deslorelin in Dogs: Six Cases (2005–2022)

**DOI:** 10.3390/ani13020265

**Published:** 2023-01-12

**Authors:** Stefano Romagnoli, Alice Diana, Lluis Ferré-Dolcet, Christelle Fontaine, Chiara Milani

**Affiliations:** 1Department of Animal Medicine, Production and Health, University of Padua, 35020 Padua, Italy; 2San Marco Veterinary Clinic, 35030 Padua, Italy; 3Virbac Group, 06510 Carros, France

**Keywords:** benign prostatic hyperplasia, chronic treatment, control of reproductive behavior, deslorelin, dog, urinary incontinence

## Abstract

**Simple Summary:**

This paper presents six different cases of dogs treated repeatedly with deslorelin, a drug responsible for a six- or twelve-month block of the reproductive system which is registered for use in adult male dogs and ferrets, male cats and prepuberal bitches. Efficacy and safety as well as return to fertility of such non-surgical neutering methods are well known following a single use but little if any data is available on prolonged use. The dogs presented in this paper were treated for 2 to 9 consecutive years for ensuring failure to reproduce (one case) as well as for conditions which are not found on the drug leaflet (benign prostatic hypertrophy and perineal gland disease (one case each) and control of reproductive behavior in male dogs and urinary incontinence in spayed adult bitches (two cases each). All animals were in good health during treatment and presented no short-term side effects. Flare-up reactions (an increase in reproductive behavior for 1–3 weeks after treatment) were observed in 1/4 intact males and were not observed in the spayed incontinent bitches. Deslorelin was effective in all treated dogs. Fertility was immediately regained in one male dog who sired a litter when his owner forgot to come back for re-treatment at the right time. Deslorelin implants can be considered as a safe alternative to surgical castration in specific pathologies mediated by reproductive hormones in situations in which surgical castration is not an option such as animals suffering from cardiovascular conditions or other systemic diseases making anesthesia unsafe.

**Abstract:**

Deslorelin is currently registered for the induction of temporary infertility in male dogs, male cats, male ferrets, and also prepubertal female dogs, but research has shown its usefulness for other conditions requiring chronic treatment. This paper presents six cases of dogs chronically treated with deslorelin for indications such as benign prostatic hyperplasia, control of fertility, abnormal reproductive behavior and urinary incontinence. All animals were in good health during treatment. Treatment duration was 2–9 years. No short-term side effects were observed except for flare-up reactions, which were observed only in 1/4 intact males. Two dogs developed a neoplasia: a spayed bitch treated for urinary incontinence developed a pituitary carcinoma, and an intact male dog implanted for control of fertility developed a bladder carcinoma. While the pituitary carcinoma seems unlikely to be related to deslorelin, the bladder carcinoma could be due to the neutered condition of the dog (which was treated for 9 years) as urinary tract neoplasia is more common in dogs following gonadectomy. Chronic treatment with deslorelin is regarded as safe when an animal is being treated for life. The possibility that a pause in the treatment might be helpful for the animal should be investigated.

## 1. Introduction

Controlling reproduction in small animals is becoming increasingly important both for pets as well as breeding animals. Historically, reproduction control has been achieved using progestogens until early in this century when new and challenging options became available, such as the long-acting agonists of gonadotropin-releasing hormone (GnRH). Progestogens have been largely misused due to poor attention to scientific evidence, as shown by the growing number of case reports documenting side effects due to excessive dosing in dogs [1], which have produced a generalized fear about their use.

Long-acting GnRH agonists initially cause a pituitary stimulation leading to the release of luteinizing hormone (LH) and follicle-stimulating hormone (FSH), followed by increased production of the respective gonadal hormones. Such continued release may cause a temporary increase in reproductive function (called flare-up reaction), which is observed in females [2] as induction of estrus and may become manifest in males as increased libido. This initial phase (which is often silent) is then followed by a downregulation of pituitary GnRH receptors with a long-lasting block of pituitary function, which interrupts all reproductive functions. There is currently one single long-acting GnRH agonist approved for veterinary use in the European Union: deslorelin. The other long-acting GnRH agonist, azagly-nafarelin [3], despite successful application, has never been launched and is now withdrawn from use in the European Union. The former has been the object of relevant clinical research activity internationally since the early 2000s [4,5,6,7,8]. In August 2005, the University of Padova was granted permission by the Italian Ministry of Health to import 10 samples of a deslorelin-based drug (Suprelorin™, Peptech Animal Health, Macquarie Park, New South Wales, Australia), which was marketed at that time in Australia and New Zealand as a veterinary compound for use in male dogs (document number DGVA-III/29534IP dated 29 august 2005). The request for import had been made to study potential clinical applications of deslorelin for the treatment of benign prostatic hyperplasia. In 2007, deslorelin (Suprelorin™, Virbac, Carros, France) started being distributed on the European market with an indication for induction of temporary infertility in male dogs. Soon after its launch, it became evident that deslorelin was potentially effective for several other indications, such as urinary incontinence in spayed bitches, as well as for the control of reproductive behavior in male dogs. As these conditions often require a chronic treatment, we started using deslorelin chronically in selected canine patients with the above presenting complaints. This paper reports on the chronic use of the 4.7 mg and 9.4 mg deslorelin implants in dogs treated at the University of Padova for a variety of different indications.

## 2. Materials and Methods

Clinical records of canine patients being seen as first-opinion or referral cases at the Veterinary Teaching Hospital (VTH) of the University of Padova, Italy (UNIPD) were searched using deslorelin as a keyword. The records of patients being treated for at least 2 years (4 or 2 consecutive treatments with the 4.7 mg or 9.4 mg deslorelin implant, respectively) or longer were selected and case histories as well as results of hematobiochemical, urinalysis, diagnostic imaging, histopathology and necropsy exams were collected and analyzed. Animal owners were contacted by telephone and the history of each case was reviewed, cross-checking the time sequence of events and the accuracy of the data reported in clinical records.

## 3. Results

The following cases of six different dogs treated for ≥2 consecutive years with deslorelin were retrieved. The clinical conditions for which chronic deslorelin treatment was used were benign prostatic hyperplasia (1), urinary incontinence in spayed bitches (2) the control of fertility (1) and control of abnormal reproductive behavior in intact male dogs (2). All owners had given their informed consent prior to the use of deslorelin in their animals. The six cases are grouped based on the indications for their treatment with deslorelin, and each case is briefly commented at the end.


**Benign prostatic hyperplasia in intact male dogs**


Benign prostatic hyperplasia (BPH) is a condition characterized by the abnormal growth and glandular hyperplasia in the prostate of intact male adult dogs. It’s prevalence increases as the dog ages, with >90% of intact dogs at 8 years of age showing signs of BPH [9]. Prostatic growth and secretion are modulated by 5-alpha-dihydrotestosterone; a shift in the balance between rate of production and removal of this steroid in the gland may increase the number of prostatic cells, determining BPH [10]. This may result in the formation of cysts accumulating urine, blood and prostatic fluid, which predisposes the prostate to develop infection from bacteria ascending the urethra or from blood circulation. The diagnosis of BPH is based on a demonstration of an enlarged prostate in the absence of any other reproductive or general abnormalities; the assay of canine prostatic specific esterase (CPSE), a marker for prostatic growth, is helpful in confirming the diagnosis. Registered treatments for canine BPH are antiandrogens such as osaterone acetate or progestogens such as delmadinone acetate [11]. However, for safety or practical reasons, it may occasionally be preferable to use deslorelin, particularly for chronic use [12]. Deslorelin should not be used in complicated BPH cases in which prostatic enlargement is causing colonic impaction or dysuria [13]. Dogs of reproductive value may also be treated (off label) with a competitive inhibitor of steroid 5alpha-reductase such as finasteride or flutamide [10].

### 3.1. Case 1. Benign Prostatic Hyperplasia in an Intact German Shepherd Dog

An 8-year-old, 33.7 kg body weight (BW), intact male German shepherd dog of the Arma dei Carabinieri was referred to the VTH of UNIPD as an emergency case for difficult defecation and ear problems, which had led to a reluctance to perform his duties as an anti-drug dog.

On the day of referral, the dog’s general conditions were normal, although he was underweight due to poor appetite, had normal skin (except for the left ear), normal subcutis and mucosal membranes, normal lymph nodes, and showed local alopecia and purulent ear discharge from his left ear. Abdominal palpation was unremarkable. On ultrasound, the dog’s prostate was increased in size and a few small cysts and a large (3.1 × 3.2 cm diameter) prostatic cyst were observed (Figure 1).

Benign prostatic hyperplasia (BPH) and purulent otitis were diagnosed, and an antibiotic treatment was prescribed. Two weeks after treatment, the dog’s conductor reported that the dog’s skin and ear condition as well as his ability to work had improved but appetite was still poor and defecation was still difficult. The prostate was not painful on palpation. A blood sample was collected for a complete blood count (CBC) and serum biochemistry evaluation. CBC revealed mild eosinophilia while biochemical parameters were normal except for high total proteins (88 g/L) and high albumin (36 g/L). Urinalysis revealed the presence of erythrocytes (250 cell/L), leucocytes (25 cell/L) and proteins (25 mg/dL) in urine. On prostatic ultrasound, the prostate was regarded as similar to the previous exam (Figure 2).

A 4.7 mg deslorelin implant (Suprelorin™, Virbac, Carros, France) was administered in the subcutaneous tissue between the shoulder blades. 

The dog returned for monthly follow-ups to monitor the implant’s action during the first 6 months, at which times a clinical exam, which included a prostatic ultrasound and a gonadotropin-releasing hormone (GnRH) stimulation test, was performed. The dog’s conditions improved remarkably already at the first monthly check when the dog showed no more difficult defecation and a normal appetite, and the ultrasound showed a progressive reduction in prostatic size and disappearance of all prostatic cysts. Upon reimplantation in March 2006, the dog’s prostate was very small (2.9 × 2.8 cm) and showed a normal texture (Figure 3).

Figure 4 shows changes in testicular size as well as testosterone concentration during the 6 months follow-up (October 2005–March 2006) after the first deslorelin treatment. Administration of the 4.7 mg implant was repeated three times in October 2005, March 2006 and September 2006, while in March 2007 a 9.4 mg deslorelin implant was administered. The dog continued to work normally throughout these years and never lost his ability to detect drugs. He was retired from his duties in 2008 and adopted from a local family we had no contact with. He lived well until his death for natural reasons in 2010.

Comment—The active principles suggested for BPH treatment are inhibitors of steroid 5alpha reductase such as finasteride and flutamide, or molecules such as osaterone acetate (not yet marketed at that time) or delmadinone acetate which compete with androgens for their receptors. This case shows that deslorelin may be used as a treatment for BPH in dogs for which fertility is not a priority or is not immediately required. Following the removal of testosterone from the general circulation, the prostate quickly decreases in dimensions reaching by the end of treatment a size comparable to that of the prostate of a surgically castrated dog. In addition, the prostatic cysts disappeared completely within one month from implantation. However, due to the flare-up phase, a worsening of the clinical situation may be observed when deslorelin is used as a first line treatment for BPH. This medical option should be avoided if the prostate is inflamed or extremely enlarged. The duration of action of deslorelin, or rather the relevant frequency of implant administration in case of BPH, is difficult to determine as prostatic growth is a slow process: once the prostate has become almost atrophic at the end of a deslorelin treatment it may take weeks or a few months before it starts giving problems due to BPH. 

The GnRH simulation test was performed in order to assess the status of the hypophyseal–pituitary–gonadal (HPG) axis as this was the first case in which deslorelin was ever used at our institution; such a test is not usually necessary in practice when using deslorelin. The assay of CPSE was not yet marketed at the time this dog was examined. 


**Urinary incontinence following spaying**


Urinary incontinence (UI) is the involuntary leakage of urine, typically occurring during recumbence or standing [14]. The problem may occur in dogs of any age or sex; however, 75% of adult cases are reported in spayed females, mostly of large sized breeds [15]. This is due to the most common cause of acquired UI being urethral sphincter mechanism incompetence (USMI), where after spaying the sphincter is weakened leading to reduced urethral closure [16]. Although in the past USMI was attributed to loss of estrogens and a consequent reduced sympathetic tone of the urethral sphincter, more recent studies attribute the incompetence to elevations of GnRH and LH (due to absent gonadal feedback) leading to decreased contractility of smooth muscle in the lower urinary tract [17]. The incidence of post-spaying USMI is between 5% and 20% in bitches. Medical treatment of UI due to USMI is normally achieved using steroids or sympathomimetic drugs, individually or combined, to take advantage of the estrogens’ action in increasing the expression of α-agonist receptors [14,18]. Treatment with GnRH agonists is a more recent alternative that is proving to be useful in approximately 50% of cases [19].

### 3.2. Case 2. Urinary Incontinence in a Spayed Boxer Bitch

A 4-year-old, 35 kg BW, spayed Boxer bitch was referred to the VTH of UNIPD in early 2002 with a presenting complaint of urinary incontinence following spaying. The bitch was ovariohysterectomized at 2 years of age. Two weeks after spaying, the owner started to find small amounts of urine on the floor. The bitch was initially successfully treated by the referring veterinarian with a human drug Temporinolo™ (Sanofi, Paris, France) SID (35 mg phenylpropanolamine + 8 mg chlorphenamine) and later referred to the VTH at UNIPD when the drug was taken off the market. 

On clinical examination on the day of referral, the bitch was healthy and in good body conditions, and all her clinical parameters as well as hematology and serum biochemistry were normal. Abdominal ultrasonography showed a normal appearance of the urinary system. A neurological exam revealed no abnormalities. Based on the dog’s history and absence of any clinical abnormality, a diagnosis of urinary incontinence (UI) due to urethral sphincter mechanism incompetence (USMI) was tentatively made (pending urinanalysis results). A 1 mg/day dose of estriol (Ovestin™, Organon, Oss, The Netherlands, the veterinary product was not yet available at that time) was prescribed and, after some attempts at tapering down the dosage, a 0.5 mg/day dosage administered early in the morning proved successful. Over the following 3 years, the bitch was re-examined annually. She remained clinically normal, continent, presented no side effects and her hematology, serum biochemistry and urinalysis remained normal.

Three years after the start of estriol administration, deslorelin was suggested as an alternative. In November 2005, a 4.7 mg deslorelin implant (Suprelorin™, Virbac, Carros, France) was administered in the subcutaneous tissue between the shoulder blades. The owners reported that urinary continence was restored 2 days after implant administration. The estriol protocol was suspended 2 months after deslorelin treatment without any relapse of signs of UI. The bitch remained continent without additional treatments for a total of 6.5 months (except for a single episode of UI occurring at 4 months when her family moved, which required a few days of estriol administration) and a new deslorelin implant was applied. 

The second 4.7 mg implant (June 2006) restored urinary continence, with the exception of an important episode of incontinence in correspondence with the return of the bitch’s owners from a 3-day trip. The owner administered estriol again for the following month, until the signs of UI subsided. During the medical follow-ups in the six months of efficacy of the implant, the owner reported normal eating and drinking behavior, good appetite and normal interaction/playing behavior. 

In December 2006, a third 4.7 mg deslorelin implant was applied. Upon a clinical and blood check prior to the following deslorelin administration the bitch showed normal hematology and an increase in Creatine Kinase (CK), cholesterol and albumin on serum biochemistry.

A fourth 4.7 mg deslorelin implant was applied in June 2007. Multiple incontinence episodes occurred 1 month after this implant, but they were controlled with estriol. The owner reported overall normal behavior, except for an increase in water intake.

The next implant was scheduled for December, but the owner delayed the procedure by a month and during this time more involuntary micturition episodes occurred. In January, a 9.4 mg deslorelin implant was suggested to extend the efficacy window of the treatment to 12 months. During the clinical follow-up scheduled 3 months after the 9.4 mg deslorelin implant insertion, the owner reported that the bitch drunk about 5 L of water a day, particularly at night, had frequent episodes of voluntary micturition of transparent urine, and showed increased appetite, pica and coprophagy. In addition, she was lazier, and unwilling to take walks or engage in other activities. A clinical follow-up was scheduled for the following month when the bitch still manifested a sluggish and depressed attitude and extreme muscle fatigue. Polyuria and polydipsia were persistent and her appetite had decreased. Episodes of epistaxis and labored breathing had also occurred. The bitch’s conditions continued to worsen and a month later she died suddenly during the night. Upon necropsy, a Cushing’s syndrome was diagnosed due to a pituitary carcinoma.

Comment—In this case, post-spaying USMI was successfully treated both with a human estriol-based compound (the estriol veterinary formulation was not commercially available at that time) as well as with repeated deslorelin implants. Urinary continence was restored about 2 days after implant administration. Despite minor variations, the duration of effect of each implant in this bitch reflected the 6-month period of efficacy in male dogs. Major stressful events appeared to impair the treatment in this bitch, as it was shown with the decrease in the implant’s efficacy shortly after the bitch’s family moved or during her owner’s absence.

Pituitary carcinomas are rare, slow-growing tumors mostly reported in older dogs [20,21], which tend to develop from the corticotroph cell line, therefore inducing hypercortisolism [22]. Hypercortisolemia was unfortunately not investigated in this bitch; however, her clinical signs of polyuria/polydipsia are clear indications of a derangement of pituitary corticotroph function. Corticotroph carcinomas and pluri-hormonal pituitary adenomas and carcinomas (where a single tumor consists of two different hormone-secreting cell types and therefore expresses multiple hormones) have been reported in two dogs and three cats [22] with invasive pituitary neoplasms composing only 11.5% of secondary intracranial neoplasms in dogs. The complex tumoral condition encountered upon necropsy, with multiple coexisting neoplasms, is often encountered in Boxers, where intracranial neoplasia is especially prevalent [23]. The pathogenesis and cause of these tumors is still unknown, yet their development could theoretically be related to increased positive feedback signals or decreased negative feedback signals relatively to one of the different hormone-secreting cell types [22]. The possibility that prolonged downregulation of pituitary GnRH receptors caused by deslorelin favored the development of a pituitary neoplasia in this dog deserves attention. However, it should be underlined that the target of deslorelin action is the gonadotroph and not the corticotroph cells and that a correlation between treatment with deslorelin and the development of pituitary carcinomas has never been reported. However, deslorelin has not been on the market for too long and the few cases in which it has been used chronically do not allow us to rule out the existence of links between the prolonged downregulation of pituitary gonadotroph receptors and the development of pituitary neoplasia. Reimplanting with deslorelin after more than 6 (or 12, depending on the implant) months following the previous implant causes the occurrence of the initial period of gonadotropin release, which is due to pituitary stimulation by deslorelin, something which in this bitch occurred twice. The effect of the repeated flare-ups between the prolonged periods of downregulation should be further investigated.

### 3.3. Case 3. Urinary Incontinence in a Spayed Rhodesian Ridgeback

A 3-year-old, 36.5 kg BW, spayed Rhodesian Ridgeback bitch was referred to the VTH at UNIPD with a presenting complaint of UI. The bitch was ovariohysterectomized at 2 years of age. Two months later, the dog began presenting signs of UI. A general physical examination showed that the patient was healthy, with a palpable abdomen and normal clinical parameters. A sample of urine was collected via spontaneous micturition to rule out cystitis, urine biochemistry was unremarkable and the urinary protein and urinary creatinine ratio was 0.02. The incontinence continued untreated for 6 months before examining the bitch. The owner reported that the leaking episodes occurred with variable incidence, both during recumbency and during movement without voluntary signs of micturition. A blood sample was collected for a CBC and serum biochemistry evaluation. CBC was unremarkable, while blood chemistry revealed high alanine aminotransferase (ALT = 424 U/L), high globulins (49 g/L) and high total proteins (80.67 g/L). Urinalysis was unremarkable and urinary creatinine ratio was 0.03. An abdominal ultrasound evaluation revealed no abnormalities. 

A diagnosis of USMI was made and a pharmacological therapy with estriol (Incurin™, MSD, Rathway, NJ, USA) 1.0 mg SID was started. Urinary continence was restored for 3 months, after which the patient started leaking urine again. The signs of UI only lasted for the summer months and disappeared during fall. At the start of winter, UI began occurring more than once a day; therefore, the bitch was seen again. Clinical examination revealed that the dog was in health and her clinical parameters were normal. Her abdomen was palpable. Body weight was 35.2 kg. The vulva appeared to be slightly enlarged, vulvar mucosa was normal and a drop of urine was visible. A vaginal smear was performed, which revealed low cellularity, the presence of non-keratinized cells and moderate neutrophilia. Urinalysis was unremarkable and urine culture showed no bacterial growth. Estriol treatment was withdrawn and substituted with phenylpropanolamine (Propalin™, Vétoquinol, Magny-Vernois, France). Urinary continence was restored for 6 months, after which signs of UI recurred and the treatment was suspended. At this time (July 2020), deslorelin was suggested as a possible treatment, and a 4.7 mg deslorelin implant (Suprelorin™, Virbac, Carros, France) was administered in the subcutaneous tissue between the shoulder blades. The treatment was effective for about 19 weeks. During the 20th week, signs of UI recurred and the owner reported occasional tenesmus, a prolonged defecating position and soft feces. The dog’s general health was normal. Rectal exploration revealed the presence of formed feces and presence of a bloody-purulent secretion from the left perianal gland, which was manually expressed. An antibiotic therapy of metronidazole-spiramycin (Spiroxan™, Ceva, Libourne, France) associated with the non-steroidal anti-inflammatory drug meloxicam (Meloxoral™, Dechra, Northwich, England, UK) 1 mg/kg SID for 7 days was prescribed to treat the anal sacculitis, which was suspected to be caused by the chronically softened feces. A 9.4 mg deslorelin implant was applied (December 2020). Eight months later, the bitch was seen again because the owner had to move and wanted to treat the dog again even though the effect of deslorelin had not disappeared yet. The bitch appeared to be in good general conditions. The owner reported that sporadic episodes of tenesmus had occurred over the previous days. A general physical examination showed normal clinical parameters (pulse, respiration and rectal temperature, auscultation, abdomen palpation, skin and subcutis). Body weight was 34.5 kg. A new 9.4 mg deslorelin implant was administered (August 2021) in the subcutaneous tissue between the shoulder blades. A blood sample was collected for a CBC and serum biochemistry evaluation. CBC revealed low platelet count (186 10^3^/μL); blood biochemistry reported low calcium (Ca = 8.57 mg/dL), low glycemia (77 mg/dL) and low azotemia (20 mg/dL). Serum protein electrophoresis revealed a low portion of albumins and high portion of α1-proteins. Urinalysis was unremarkable and urine culture showed no bacterial growth. Based on a follow-up call 8 months after administration of the last implant, the bitch was in good health and lively, her weight had increased to 37.5 kg and she had remained continent throughout the time.

Comment—Post-spaying USMI in this bitch was treated with estriol and phenylpropanolamine with some success, although relapses were noticed following prolonged use of both drugs. Such “delayed” treatment failures are occasionally observed in incontinent bitches treated with both the above drugs. Deslorelin administration in this bitch has proven effective in the long run and the treatment is currently being successfully continued without any negative side effects. Treatment with deslorelin allowed the signs of UI to disappear very quickly within 1–3 days after implant administration. The short duration of effect of the 4.7 mg implant in this bitch might have been due to the inflammatory condition, which had developed on her anal glands causing tenesmus thereby presumably increasing pelvic contractility and bladder instability. In this patient, the duration of effect of the 9.4 mg implant for the management of urinary incontinence seems to be at least 8 months (as the second one was administered when the first one was still active), if not longer. More cases however are needed in order to confirm this observation. Variations in body weight appear to be unrelated to use of deslorelin but rather due to the bitch perhaps being stressed by having to move frequently with her owner between Padova and Rome; once she settled down in Rome during the last year her weight went back to normal.


**Controlling fertility and abnormal reproductive behavior in male dogs**


The control of fertility as well as abnormal reproductive behavior is a common presenting complaint in male dogs as owners may be concerned about their male dogs mismating estrous bitches as well as displaying strong libido and mounting, inter-male aggressiveness, aggressive dominance towards owners and roaming behavior [24,25]. Male reproductive behavior is dependent on serum testosterone concentration and is influenced by the animal’s experience during growth and development; therefore, male behavioral conditions that are thought to be due to testosterone secretion can be treated with castration or the administration of antiandrogens or GnRH-agonists as well as behavioral training [26,27]. A lack of development of most of the above features of male reproductive behavior occurs when males are castrated prior to puberty, while if castration is performed following puberty and particularly when a male has been exposed to or has even bred female/s in heat its effectiveness in controlling annoying male behaviors is greatly reduced. GnRH-agonists downregulate the HPG axis thereby decreasing serum T concentration [7,8]; their use has proven effective in decreasing all aspects of male reproductive behavior except for aggressiveness—which has a dual connotation (hormonal and behavioral) and therefore cannot always be controlled by either surgical or medical neutering. 

### 3.4. Case 4. Controlling Fertility in a Male Maremma Shepherd Dog

A 2-year-old, intact, 34.5 kg BW, male Maremma shepherd was presented for pharmacological neutering due to a history of breeding with his sister producing a litter of eight normal pups. His vaccination and heartworm prevention programs were current and the dog did not have any health issues. 

On the day of referral all clinical parameters were normal including prostatic size on rectal palpation. A blood sample was collected for a complete hematological evaluation and testosterone levels, with results being unremarkable. In March 2006, a 4.7 mg deslorelin implant (Suprelorin™, Virbac, Carros, France) was administered in the subcutaneous tissue between the shoulder blades. Six months later, the dog was clinically normal, his testicles had a soft consistency and a size (measured with a caliper) of 2.3 × 1.5 cm (right) and 3 × 1.2 cm (left) and serum testosterone concentration after GnRH stimulation was <0.2 ng/mL. In September 2006, the dog was implanted with another 4.7 mg deslorelin implant.

The dog was then rechecked three times at 2-monthly intervals and was found to be always in good general conditions, active, lively and with good appetite. In May 2007 (8 months following the previous treatment), his testicles were still soft in consistency and with a size (measured with a caliper) of 2.7 × 1.5 cm (right) and 2.9 × 1.4 cm (left) and serum testosterone concentration was still <0.2 ng/mL; the dog was treated with a third 4.7 mg deslorelin implant.

In February 2008 (8.5 months later), the dog was seen again and this time a 9.4 mg implant was administered. This treatment was repeated in May 2009, then in May 2010 the dog was seen again for another deslorelin treatment. At this time, the dog’s owners asked for the dog to be treated with the 4.7 mg implant as they were convinced that the dog was less lively and active and with a stronger tendency to gain fat when treated with the 9.4 mg implant. 

From this time onward, the dog was treated regularly every 6 months for the following 5 years (always with the 4.7 mg implant) during which time he remained in normal clinical conditions based on regular clinical checks. The timely sequence of all deslorelin treatments and measures of each testis due to the effect of the different deslorelin implants are summarized in Table 1.

Testis consistency remained soft and with a good mobility. The prostate was also small and normal, except for one ultrasonographic examination in November 2010, which revealed a mild increase in volume and hyperplasia, cranial repositioning and slight asymmetry compared to previous measurements. The prostate returned to a normal appearance after a new implant.

The 4.7 mg implant of February 2013 was delayed as the owner forgot about it and the dog was treated a month later on 20 March 2013. At this time, testicular size (right testis 4.2 × 2.2 cm, left testis 4.2 × 2.0 cm) and normal consistency indicated that the dog had presumably regained full fertility by the time he was implanted. The prostate remained normal in size, consistency and symmetry. The dog’s sister came in heat less than a month later and breeding between the two dogs resulted in pregnancy with birth of a litter of three pups.

Since 2014, the implant was placed in the subcutaneous tissue of the periumbilical area. The dog remained healthy and with small, soft testes and a small prostate during the 2 years following February 2014. During the Spring 2015, he developed dysuria, hematuria and difficult defecation. Survey abdominal X-ray and ultrasonography showed the presence of a mass at the level of the bladder neck. A computerized tomography exam showed the mass to be originating from the inner aspect of the bladder. Surgical removal was attempted but only a biopsy was made as the mass had already involved both ureters. A histological diagnosis of bladder adenocarcinoma was made, then the dog was treated with chemotherapy for a few months and euthanized in December 2016.

Comment—This case shows that deslorelin is effective in maintaining male dogs in a permanent state of sterility and fertility can be easily regained as soon as the treatment is discontinued. Semen quality following the 6 month treatment duration is reported to be restored between 3 and 5 months after implant removal [28]. This seems to be the case for most deslorelin treated dogs in which the implant is not removed although in a few cases a dog may take up to 1 or more years to regain full fertility [29]. Repeated deslorelin implant administration in this dog caused prolonged sterility. Testicular volume could not always be determined accurately in this dog as his owner would commonly refuse testicular ultrasonography on financial grounds; however, testicular size and consistency was reduced during treatment based on clinical assessment and increased quickly as soon as treatment was discontinued. Deslorelin treatment allowed to maintain this dog in a continuous sterile condition for 7 years and, as soon as the owner forgot to bring him back for a treatment, he bred his sister again and three pups were born. Except for this short gap in 2013, the dog was treated for a total of 9 years and his general health remained normal throughout this period. The incidence of bladder carcinoma in dogs is low (about 2% of all canine tumors) [30]. Some breeds such as Scottish terriers, Shetland sheepdogs and West Highland white terriers are at higher risk [31]; however, there is no information on the incidence of this condition in Maremma shepherds. Incidence is reported to be affected by neutering [31,32]. The possibility that prolonged chemical neutering caused by deslorelin acted similarly to surgical neutering in favoring the development of bladder neoplasia in this dog cannot be ruled out and should be taken into consideration. The duration of effect of the 4.7 mg implant in this dog was frequently longer than 6 months while duration of the 9.4 mg implant could not be assessed as the dog was reimplanted always at the end of the 12th month. The belief that the 9.4 mg implant was causing the dog to be less active and gain more weight is not supported by any objective fact. Such a complaint has never been reported to the VTH of UNIPD by any other owner of dogs being treated with the 9.4 mg deslorelin implant. In this dog, no side effects directly due to deslorelin and no flare-up reaction following any treatment were ever observed.

### 3.5. Case 5. Controlling Hypersexuality in an American Staffordshire Male Dog

A 2-year-old, 23.5 kg BW, male, intact American Staffordshire dog was referred to the VTH of UNIPD for hypersexuality and benign prostatic hyperplasia. Soon after puberty, he had started showing an excessive amount of energy in all his behaviors both at home with continuous mounting of the legs of his owners followed by ejaculation as well as when taken out with aggressiveness against other dogs and a continuous pulling very strongly on his leash, which made him very difficult to control. The dog also had pain in his lower abdomen, perianal gland pain and would occasionally show hematuria at the beginning of spontaneous micturition. Owners also reported that feces were ribbon-like and pasty, with a final bloody spray a few days after stimulation from females in heat. 

Upon referral (May 2015), clinical examination revealed that all clinical parameters were normal except for his perianal glands, which were slightly swollen and painful and his prostate, which on ultrasound appeared increased in size and with one large 3 × 2 cm diameter cyst in the right lobe and a few smaller cysts on both lobes. CBC was unremarkable, while blood chemistry revealed high magnesium (Mg = 2.51 mg/dL), high ALT (115 U/L), high aspartate aminotransferase (AST = 49 U/L), high globulins (45 g/L) and high total proteins (74.6 g/L). Basal testosterone concentration was 5.17 ng/mL. Since his owners were very keen in avoiding surgical castration and because of the dog’s high liver enzymes, it was decided to try a treatment course with a short acting drug such as finasteride 0.5 mg/day orally (Finasteride Teva Generics, Teva, Tel Aviv, Israel), and then to re-evaluate the dog after one month to consider the option of using deslorelin. At the following check-up, one month later, the dog was healthy and in good body conditions, he was not showing perianal gland pain and only a single episode of blood in urine had been noticed by his owners. However, the prostatic cysts were still present. An additional 15 days of finasteride treatment (same protocol as above), paired with 11.5 μg/kg megestrol acetate (Estropill, MSD, Rathway, NJ, USA) were prescribed (megestrol acetate is a short-acting compound and its very low dosage makes it a safe option for dogs with questionable liver function as the treatment can be stopped at any time). At this time the dog owners declined to retest serum biochemistry on financial grounds.

The following month (July 2015), the dog was examined once more, all his clinical parameters were normal and owners reported that body functions were also normal again, with normal colored urine and well-formed feces. Perianal glands did not appear to be swollen or painful. Upon ultrasonography, the dog’s prostate appeared to have decreased in size, from 5 cm in the last measurement to 3.5 cm, and there were fewer cysts, with only 2 medium-large-sized cysts (1.8 × 1.3 cm). Again the dog owners declined to retest serum biochemistry on financial grounds. Therefore, the dog was administered a 4.7 mg deslorelin implant together with a 50 mg oral dose SID of cyproterone acetate (Androcur™, Bayer, Leverkusen, Germany) during the initial 2 weeks following implantation in an effort to avoid a worsening of prostatic conditions due to the flare-up reaction. The dog had a worsening of his behavior for about 3 weeks but then gradually improved and became a very easy-to-handle, calm and affectionate dog. 

In December, the owners reported that the dog had greatly improved: he had stopped mounting other bitches and was only playing with them, had normal urine and feces and only showed a very mild irritation of the perianal gland. All his clinical parameters were normal and US confirmed that the prostate had improved and measured 3 × 4 cm with a single cyst in the left lobe measuring 0.5 cm (>2 cm 6 months prior). At this time, the dog was administered another 4.7 mg deslorelin implant.

In May 2016, the 4.7 mg implant was repeated after confirming general health, normal behavior and the normal ultrasonographic appearance of the prostate. The dog had had a gastrointestinal problem during the summer for which he had been admitted to a veterinary clinic for 2 days, at which time his blood (hematology and biochemistry) as well as ultrasonography tests were normal. The dog recovered uneventfully. At the following appointment the owners came in late in December 2016 and by that time the dog was already showing an increase in testicular size and recurrence of the perianal gland problem. This was considered to be due to the loss of the effect of the previous implant and, when a new 4.7 mg deslorelin implant was administered, the dog showed a 3-week flare-up reaction (Table 2). Four months later, a 9.4 mg deslorelin implant was voluntarily administered in advance to avoid any risk of a new flare-up. From this time onward, the 9.4 mg deslorelin treatment was repeated yearly.

A year later (March 2018), clinical examination revealed all the dog’s clinical parameters were normal. A new 9.4 mg deslorelin implant was administered in the periumbilical region.

The following year, the implant was administered in April and only remained effective until November, when testicles increased in size, frequent ejaculations occurred and the dog’s behavior became erratic and very agitated again. For this reason, the following 9.4 mg implant was administered before the 12th month, in December 2019. This implant’s effectiveness was also shorter, with renewed problems beginning in June 2020: increased testicular size, frequent ejaculations and mounting of his owners. 

In October 2020, the 9.4 mg implant was repeated after confirming general health and a US showing increased prostate size and two large cysts. Another 9.4 mg implant was administered 11 months later in September 2021.

Comment—This dog is very unusual because of his extreme vigor, libido and restlessness, which make him a difficult dog to manage. His behavior is probably the result of a very strong testosterone production or an alteration of its receptors, which is also causing his prostatic and perianal gland problems. This dog would obviously benefit from removal of his gonads as demonstrated by his improvement during periods in which he was being treated with deslorelin. In a case like this, it is fundamental to avoid a flare-up reaction. When his owners forgot to bring him back in November 2016 and treatment was delayed a few weeks, the dog’s testicles started to increase in size immediately and the dog resumed his libido-related behavior making life very difficult for his owners. In order to avoid flare-up reactions in cases like this dog, it is important to schedule an appointment prior to the end of the previous implant’s action in order to allow the dog’s gonads to remain constantly under the effect of deslorelin. The two flare-up reactions of 2019 and 2020 were unexpected. As no testosterone measurements were performed, a shorter duration of efficacy of the implant cannot be confirmed. Possible explanations could be some product defects or incorrect technique of administration. A different mechanism of action of the implant for the management of behavior other than the one involved in the control of the synthesis and release of testosterone cannot be ruled out. Interestingly, the fact that the last implants used in this dog had the expected normal duration of efficacy proves that the dog remained sensitive to the action of deslorelin. 

### 3.6. Case 6. Controlling Hypersexuality in a Mixed-Breed Male Dog

A 3-year-old, 5.8 kg BW, mixed-breed, intact male dog was referred to the VTH of UNIPD for pharmacological neutering. The dog had a history of aggressiveness towards other dogs but not humans, and a strong libido, which led him to the display of frequent masturbation and attempts to mount his owner, as well as a tendency to develop inflammation of the perianal glands. The dog lived indoors and, despite being fed dry food ad libitum, had recently lost 800 g during the previous 2 months. The owner also reported the dog as being hyperactive and having a voracious appetite and polyuria/polydipsia. On clinical examination, the dog was alert and normally responsive, his clinical parameters (pulse, respiration and rectal temperature) were normal and his testicles were normal in size, shape and consistency. Testicular size was assessed with a caliper: the right testis measured 3.24 cm^3^ (applying volume formula L × W × H × 0.71 [33]) and the left measured 3.2 cm^3^. As the dog could not be restrained manually, he was sedated with a premedication of 0.2 mg/kg of butorphanol and 3 µg/kg of dexmedetomidine to carry out clinical procedures. A blood sample was collected for a CBC and serum biochemistry evaluation, which were unremarkable except for a slight neutropenia (3870/μL) and decrease in red blood cells (RBCs) and platelet distribution width (PDW), low total proteins (59.33 g/L), high C-reactive protein (CRP = 2.3 mg/dl) and low globulins (27 g/L). Urinalysis revealed alkaline urine (pH = 8) and a modest increase in specific gravity (1.050); the urine culture showed no bacterial growth. In October 2020, deslorelin was suggested as a potential treatment and a 4.7 mg deslorelin implant was administered in the subcutaneous tissue between the shoulder blades. The clinical follow-up 3 months later (January 2021) revealed that a flare-up effect had not taken place. The owner reported that the libido had diminished, but the aggressiveness towards other dogs had not. In addition, little if any effect on the perianal glands was observed, as the dog continued licking his perineal area and scratching it by dragging his posterior on the ground. Clinical examination revealed that the dog was alert and normally responsive and his clinical parameters were normal. He had gained 0.5 kg since the last examination. Ultrasonography showed pharmacologically induced prostatic hypotrophy, since the prostate measured 1.12 cm^3^, while a volume of 5.59 cm^3^ was expected in the dog (BW = 7 kg); in addition, the right testis measured 1.28 cm^3^ (applying volume formula L × W × H × 0.71 [33]) and the left 1.74 cm^3^. 

A second follow-up occurred in March 2021, and the owner reported an improvement in appetite and perianal glad inflammation, along with a reduction in libido, with decreased leg humping and masturbation. The testes were measured with a caliper (volume was calculated as previously: the right testis measured 0.74 cm^3^ and the left testis measured 0.96 cm^3^) and ultrasonography (right testis 2.33 cm × 0.62 cm and left testis 0.8 cm × 0.69 cm). At this time, the dog weighed 7.2 kg.

The following month (April 2021) marked 6 months from the implant and the dog was examined again. The owner reported that the dog had been in good health and active, with a normal general state. In the last 10 days, 2–3 mounting episodes had occurred. Clinical examination showed good general conditions and a BW of 7.1 kg. CBC revealed mild lymphocytosis (2920/μL) and monocytopenia (270/μL). Blood biochemistry reported low total proteins (61.45 g/L), high albumins (33.93 g/L) and low globulins (28 g/L) resulting in a high albumin/globulin ratio (1.23). The dog was assessed as being presumably at the end of the function of the deslorelin implant. He was sedated with 0.2 mg/kg of butorphanol and 3 µg/kg of dexmedetomidine and a new 4.7 mg deslorelin implant was inserted.

A follow-up examination 2 months later (June 2021) showed that the dog was healthy, with no signs of flare-up effect and serum testosterone was undetectable. The dog was re-examined 5.5 months after the deslorelin implant. The owner reported an efficacy of treatment for 5 months and an increase in mounting episodes in the last 2 weeks. They initially occurred once or twice per week and then increased in frequency to daily occurrences. Upon examination, clinical parameters were normal and the testes did not appear to have increased in size. A blood sample was collected under sedation (same drugs as above), CBC revealed neutropenia (4690/μL), low platelet count (74 × 10^3^/μL) and presence of platelet aggregates, and high RBC (7.7 × 10^6^/μL), Hgb (18 g/dl) and Hct (52.7%). Biochemistry reported low total proteins (61.45 g/L), high albumins (33.93 g/L) and low globulins (28 g/L) resulting in a high albumin/globulin ratio (1.23). In September 2021, a 9.4 mg deslorelin implant was inserted in the subcutaneous tissue at the level of T12. In March 2022, on clinical examination, the dog weighed 6.8 kg and owners reported he had been well, eating with appetite and with normal general health. He had not shown interest in bitches in heat and reported few episodes of masturbation. A blood collection could be performed without sedation; CBC was unremarkable and blood chemistry parameters fell within the reference intervals. Serum testosterone following a GnRH stimulation test (carried out with 50 μg gonadorelin administered SC 1 h previously) was 0.7 ng/mL.

Comment—In this dog, repeated deslorelin treatments were effective in reducing libido and masturbation and normalizing his dominant attitude with an improvement in his relationship with his owner as well as an increase in weight and body conditions. There were no treatment-related negative effects; however, deslorelin was not effective for aggressiveness towards other dogs as this is evidently a behavioral condition requiring an appropriate behavioral approach. The 4.7 mg implant was effective in this dog for the behavior management for about 5 months and clinical signs due to strong libido would gradually recur from the start of the 5th month post-treatment, while only 7 months have elapsed since administration of the 9.4 mg implant and therefore it is too early to assess the duration of this implant in this dog. Testicular ultrasonography could not be reliably performed in this dog to calculate testis volume due to costs.

## 4. Discussion

Deslorelin should be regarded as a very safe drug. The majority of short-term side effects due to deslorelin have been reported in intact bitches (a category which was not featured in our study) such as persistent estrus, uterine disease, ovarian cysts, pseudopregnancy, UI, cystitis, increased weight and behavioral as well as coat changes [6,34,35,36]. Weight gain and a change in temperament were observed in case n. 4, the Maremma shepherd dog treated for 9 years, although such effects were specifically reported following the use of the 9.4 mg implant and not following the use of the 4.7 mg implant. This difference in weight gain and behavior between the two types of deslorelin implants is hard to explain and might be due to a coincidence with other subclinical condition/s affecting the dog. Unwanted pregnancy is also a possible side effect following a deslorelin-induced heat as such pregnancy is typically followed by a mid-term abortion in bitches [6]. The occurrence of heat is very difficult to avoid in intact post-pubertal bitches treated with deslorelin. In bitches, the administration of a deslorelin implant during diestrus will drastically reduce the incidence of heat [6] but some bitches may still develop flare-up signs and occasionally ovarian cysts and pyometra, particularly when treated during a luteal phase [37]. When compared to other drugs to achieve a long-lasting control of reproduction and reproductive behavior such as progestins, deslorelin is certainly advantageous as it achieves a block of the HPG axis following a prolonged pituitary stimulation leading to pituitary exhaustion and ultimately resulting in the absence of all pituitary-controlled reproductive hormones from the organism. Progestins instead block the HPG axis thanks to a feed-back mechanism obtained by the administration of exogenous hormones. Although progestins should be considered as safe drugs (provided that they are used only in the right patient for the correct amount of time and at an appropriately low dosage), it is undisputable that their use constitutes extra work for the liver and kidneys. Therefore, progestins have some limitations connected with the duration of treatment and health conditions of the patient. Deslorelin has no such limitations and in fact it should be safely used in animals with renal insufficiency who cannot undergo surgical neutering.

Long-term problems following the use of deslorelin have been reported in intact bitches [37] but have not been observed yet in other categories of patients (intact male dogs and spayed bitches). We observed two cases of neoplasia in our patients n. 2 and 4. The case of pituitary carcinoma developing in the Boxer bitch with urinary incontinence (case n. 2) deserves further investigation. It appears unlikely to be related directly to the use of deslorelin as it presumably developed from the adrenocorticotroph-secreting cells, which are not a target of deslorelin action. However, abnormal pituitary stimulation has been proposed as a cause of this type of tumors [22]. Deslorelin produces a prolonged rather than abnormal pituitary stimulation, which does not cause any pituitary derangement in normal, young or adult animals, although it remains to be established whether or not such a prolonged stimulation may be a problem for elderly animals, in which flare-up effects should probably be avoided. The effect of the prolonged use of GnRH agonists on the pituitary gland and consequent repeated initial stimulation should be further investigated.

Similarly, no direct connection can be established between deslorelin administration and the bladder carcinoma that developed in the Maremma Shepherd dog (case n. 4). GnRH receptors can be found in many organ systems including the urinary tract of dogs [38], and bladder function in spayed bitches is directly influenced by GnRH agonists [39]. However, the role of GnRH agonists on the urinary function of intact male dogs, if any, is unknown. Neutered dogs are at a higher risk of developing urinary tract and prostatic neoplasia [40,41,42,43]. As the Maremma shepherd dog was treated throughout most of his life with deslorelin (except for a short lap of a few months in 2013), his long-lasting (chemically) neutered condition might have played a role in the development of his bladder neoplasia. Perhaps a short pause (3–6 months or longer) in deslorelin treatment might be beneficial for patients undergoing chronic treatments, allowing gonadal hormones to exert their function in stimulating the immune system.

The flare-up reaction is an interesting phenomenon: (a) it can be useful for some clinical situations (as for estrus induction in intact females); (b) it can be irrelevant for conditions, such as benign prostatic hyperplasia, as time for reimplantation can be decided by monitoring the patient and acting when clinical or diagnostic imaging signs indicate that the condition is worsening again; or (c) it can be dangerous in the case of patients treated for excess libido such as our case n. 5. Considering the potential negative effect of gonadectomy on the immune system and general health of dogs [44], the usefulness of letting dogs undergoing chronic deslorelin treatment go through short (3–6 months or longer) periods of times without treatment should be investigated.

## 5. Conclusions

The results of this study indicate that the long-term use of deslorelin may be effective in diseases such as prostate hyperplasia, reproductive behavior disorders or urinary incontinence. In addition, according to the results from these case studies, deslorelin is a safe drug to choose for a chronic treatment, where repeated implants are administered for years of the dog’s life. The possibility of a correlation between the long-term use of deslorelin and the development of pituitary carcinomas should be further investigated. Moreover, as surgical neutering has been associated with the occurrence of urinary tract and prostatic neoplasia, the possibility exists that side effects of the prolonged absence of gonadal hormones may also occur during a chronic deslorelin treatment.

## Figures and Tables

**Figure 1 animals-13-00265-f001:**
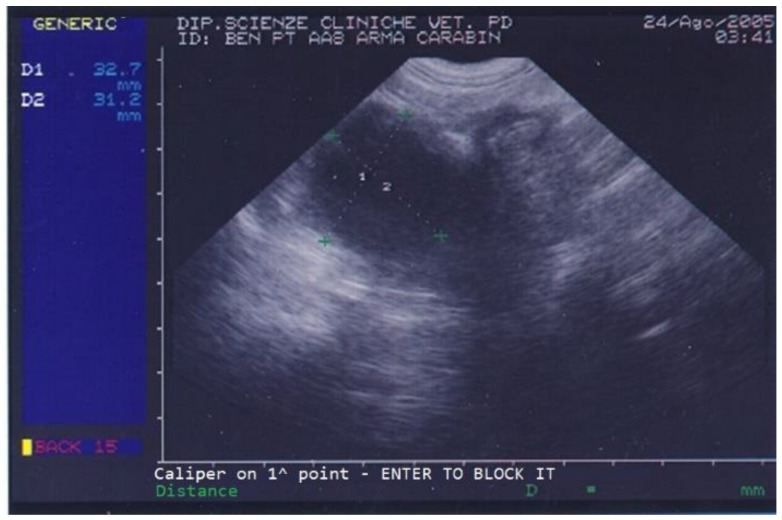
Abdominal ultrasonography of a German shepherd dog before deslorelin implant. The scan shows the ultrasonographic appearance of the prostate. The largest prostatic cysts’ perpendicular diameters are measured as D1 = 32.7 mm and D2 = 31.2 mm.

**Figure 2 animals-13-00265-f002:**
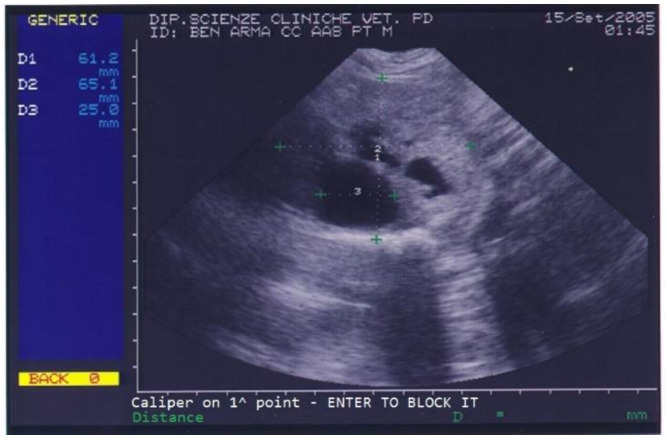
Abdominal ultrasonography of a German shepherd dog before deslorelin implant. The scan shows the ultrasonographic appearance of the prostate and the presence of different-sized cysts. The prostate’s perpendicular diameters are measured as D1 = 61.2 mm and D2 = 65.1 mm. The largest prostatic cysts’ diameter is measured as D3 = 25 mm.

**Figure 3 animals-13-00265-f003:**
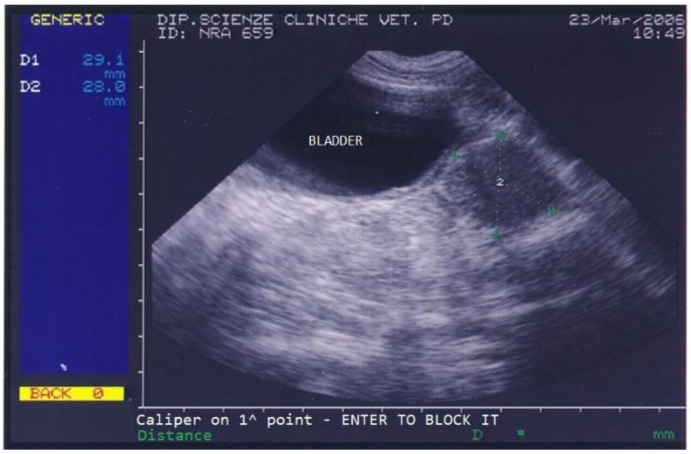
Abdominal ultrasonography of a German shepherd dog after a 4.7 mg deslorelin implant. The scan shows the ultrasonographic appearance of the bladder and prostate. The prostate’s perpendicular diameters are measured as D1 = 29.1 mm and D2 = 28 mm.

**Figure 4 animals-13-00265-f004:**
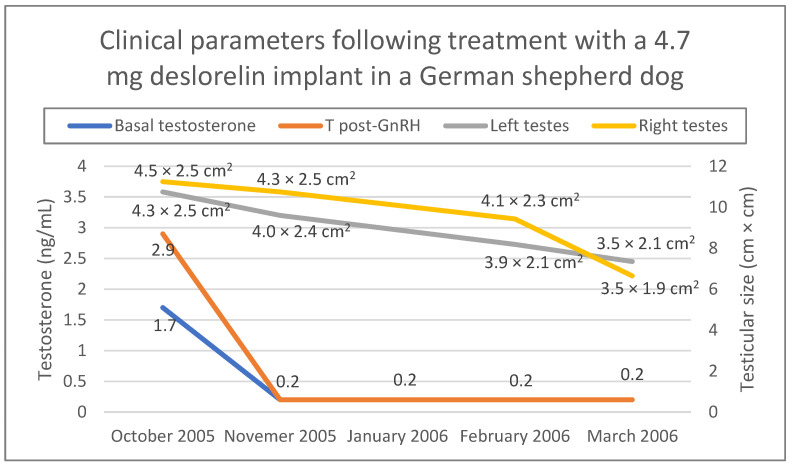
Clinical parameters (testicular size, serum testosterone concentration before and one hour after a stimulation test with 50 μg gonadorelin SC) of a German shepherd dog treated with a 4.7 mg deslorelin implant. Testosterone concentration was measured with chemiluminescence immunoassay. Testicular size was measured during ultrasonographic examination.

**Table 1 animals-13-00265-t001:** Date of implantation and variations in testicular size of a Maremma shepherd dog treated continuously with deslorelin between 2006 and 2015. Measures were taken at most clinical checks prior to reimplantation and reflect size of the right and left testes as measured during clinical examination using a stainless-steel caliper.

Date	Deslorelin Implant	Right Testicle	Left Testicle
8 February 2006	4.7 mg		
14 September 2006	4.7 mg	3 × 1.2 cm	2.3 × 1.5 cm
24 September 2007	4.7 mg	2.9 × 1.4 cm	2.7 × 1.4 cm
6 January 2008	9.4 mg	3.5 × 1.5 cm	3 × 1.5 cm
20 May 2009	9.4 mg		
22 May 2010	4.7 mg		
9 September 2010		3.2 × 2 cm	3.8 × 2.1 cm
24 November 2010		2.2 × 4 cm	2.1 × 4 cm
22 December 2010	4.7 mg	2.4 × 4 cm	2.2 × 4.1 cm
17 May 2011	4.7 mg		
23 November 2011	4.7 mg		
27 April 2012	4.7 mg		
12 October 2012	4.7 mg		
18 March 2013	4.7 mg	4.2 × 2.2 cm	4.2 × 2 cm
6 February 2014	4.7 mg		
23 July 2014	4.7 mg		
20 January 2015	4.7 mg		
23 July 2015	4.7 mg	1.8 × 3.1 cm	1.8 × 3.1 cm

**Table 2 animals-13-00265-t002:** Date of implantation, type of deslorelin implant, lot number, expiration date and presence of a flare-up reaction following treatment of an American Staffordshire intact male dog continuously treated with deslorelin between 2015 and 2021. A flare-up reaction occurred in December 2016 as the dog was reimplanted after more than 6 months following the previous 4.7 mg treatment.

Date	Deslorelin Implant	Lot Number	Expiration Date	Flare-Up Reaction
23 July 2015	4.7 mg	data	October 2016	Yes
22 December 2015	4.7 mg	SLV308B21	October 2016	No
19 May 2016	4.7 mg	SDW322D21	March 2017	No
1 December 2016	4.7 mg	SJW333F21	August 2017	Yes, 3 weeks
30 March 2017	9.4 mg	TDX353B21	March 2018	No
5 March 2018	9.4 mg	TDX353B21	March 2018	No
2 April 2019	9.4 mg	THY394C21	July 2019	No, but effect lasts until November 2019
9 December 2019	9.4mg	TGZ427D21	June 2020	Yes, 10 weeks, effects last until June 2020
6 October 2020	9.4 mg	TAB46721	December 2022	Yes, 4 weeks, treated for BPH
6 September 2021	9.4 mg	TAB46721	December 2022	No

## Data Availability

The data that support the findings of this study are available on request from stefano.romagnoli@unipd.it. The data are not publicly available due to privacy or ethical restrictions.

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
