# Peer review of "Chronic Use of Deslorelin in Dogs: Six Cases (2005–2022)"

_animals, 2023, doi:10.3390/ani13020265_

Round 1

Reviewer 1 Report

The present work from Romagnoli and coauthors with the title “Chronic use of deslorelin in dogs and cats: 7 cases (2005-2022)” wants to improve the knowledge about this precious, albeit partially unknown, clinical instrument. The present reviewers consider this clinical case study very useful from a clinical and scientific standpoint. Also, the way the clinical cases are presented stimulate discussion and considerations from the reader.

Only minor changes are suggested, written below:

Line 8: add a comma after “dogs” and after “treatment”.

Line 44: delete “albeit unfounded”.

Line 57: change “function” with “functions”.

Line 62: add a comma after “2005”.

Line 67: add a comma after “2007”.

Line 174: it should be interesting to know if the deslorelin treatment was continued until death even after retirement.

Line 286: comments about the effects of deslorelin on the development of the pituitary tumor should be more cautios, both here and in the abstracts, as there is no literature about it and, as underlined by the authors at line 193, different “flare-ups” happened during the life of this dog.

Line 359: about this comment, it should be stated and stressed that more cases are needed to interpret the different duration of the implant and confirm the opinion of the Authors.

Line 394: did you perform a stimulation test or only detected the basal testosterone?

Line 440-441: authors should specify that this is not granted 100% of the times, as specified in the leaflet.

Line 560: very interesting. Has the dog been visited also by a veterinarian specialized in behavioral problems?

Line 635: change “rather than pharmacological therapy” with “approach”.

Line 770: delete spaces between the citation numbers.

Citation n 37: delete “3” from 360 (indication of pages), or add the “3” to “53” in the citation n 38, and align the other citations in the bibliography.

Author Response

Response to Reviewer 1 Comments

Point 1: Line 8: add a comma after “dogs” and after “treatment”.

Response 1: Commas added as requested.

Point 2: Line 44: delete “albeit unfounded”.

Response 2: Deleted.

Point 3: Line 57: change “function” with “functions”.

Response 3: Changed.

Point 4: Line 62: add a comma after “2005” &  Line 67: add a comma after “2007”.

Response 4: Commas added.

Point 5: Line 174: it should be interesting to know if the deslorelin treatment was continued until death even after retirement.

Response 5: Unfortunately, after retirement the dog was no longer under our supervision and the contact with the new owners was lost, for this reason we think he was not implanted anymore.

Point 6: Line 286: comments about the effects of deslorelin on the development of the pituitary tumor should be more cautios, both here and in the abstracts, as there is no literature about it and, as underlined by the authors at line 193, different “flare-ups” happened during the life of this dog.

Response 6: We thank the reviewer for this comment, we changed this part accordingly, as to erase any possible misunderstanding on the effect of deslorelin and occurrence of pituitary carcinoma.

Point 7: Line 359: about this comment, it should be stated and stressed that more cases are needed to interpret the different duration of the implant and confirm the opinion of the Authors.

Response 7: We agree with the reviewer and we added a statement as suggested.

Point 8: Line 394: did you perform a stimulation test or only detected the basal testosterone?

Response 8: Yes, we did, apologies for overlooking this, we always do a GnRH stimulation test, we do not always collect a basal sample but we certainly inject GnRH and then draw blood one hour later. Therefore, the results of testosterone assay reported are post-GnRH.

Point 9: Line 440-441: authors should specify that this is not granted 100% of the times, as specified in the leaflet.

Response 9: We agree with the reviewer and we added a statement as suggested, also adding two more references (Stempel, 2022; Goeriche-Pesch, 2016) on this matter.

Point 10: Line 560: very interesting. Has the dog been visited also by a veterinarian specialized in behavioral problems?

Response 10: Both owners are veterinarians and had already ruled out the hypothesis that the dog’s problem might be due to his behavior.

Point 11: Line 635: change “rather than pharmacological therapy” with “approach”.

Response 11: Changed as suggested.

Point 12: Line 770: delete spaces between the citation numbers.

Response 12: Deleted as suggested.

Point 13: Citation n 37: delete “3” from 360 (indication of pages), or add the “3” to “53” in the citation n 38, and align the other citations in the bibliography.

Response 13: Modified as suggested.

Reviewer 2 Report

Dear Authors, 

Even though the topic of this paper of yours is very interesting, in my opinion the content is not suitable for publication in an high IF journal. The reason for this is the exclusively descriptive nature of this paper. It could not be defined as a 'case series' because the cases that are reported are very heterogeneous, including animals with different sex, conditions, and species as well. Even as single case reports the cases are lacking (also please note that the ultrasound images report non-English words and they should be adjusted). No conclusion can be formulated based on what you reported and I would be careful in stating that there is absolutely no correlation between the pituitary carcinoma and the chronic use of deslorelin and that the use is safe. I found each case interesting and I strongly encourage you to enrich the caseload for each group and use this casa to design future research on the chronic use of GnRH agonists, that is indeed needed. 

Best regards.

Author Response

We thank you for taking your time to review our manuscript and presenting your report.

We understand your point but admit we disagree with you in regards to the unsuitability to be published. Although heterogeneous, the cases are interesting and follow a common thread, being the long-term treatment with deslorelin.

Aiming to make our paper more coherent we have removed the case regarding the cat, we hope that our refinement of the cases may allign better with your suggestions. We have also modified our statements regarding the correlation between long-term use of deslorelin and development of pituitary carcinomas and stated it should be further investigated since development of pituitary carcinomas has never been reported.

Reviewer 3 Report

The paper is fascinating because it reports some chronic applications of deslorelin in various clinical situations. Since situations are different, the scientific value is that of case reports, but some applications are unpublished. After almost 20 years of using the drug, it is essential to report all suspected or confirmed effects. These data are critical for the safe use of the drug and for the problems related to gonadectomy and its alternatives.

Minor points are the following

1. simple summary and abstract are basically the same.

2. in case 1 please comment that GnRH was useful for cyst resolution

3. Among the side effects of prolonged GnRH treatment authors may consider inserting citations on coat changes, hip dysplasia, haematocolpus, uterine hyperplasia, and ovarian tumours.

Author Response

Response to Reviewer 3 Comments

Point 1: simple summary and abstract are basically the same

Response 1: Simple summary has been changed as suggested.

Point 2: in case 1 please comment that GnRH was useful for cyst resolution.

Response 2: We added a comment on this point as suggested.

Point 3: Among the side effects of prolonged GnRH treatment authors may consider inserting citations on coat changes, hip dysplasia, haematocolpus, uterine hyperplasia, and ovarian tumours.

Response 3: We thank the reviewer for this comment. We appreciate the importance of side effects of deslorelin.  However, the cited effects occur in intact females. We have only treated males and two ovariohysterected bitches, for which the mentioned side effects are not applicable.  We feel that dealing with side effects of deslorelin in intact bitches would bring us out of the scope of the paper which is chronic administration in males (for which the drug is officially marketed) and in spayed females suffering from urinary incontinence (for which there is an abundance of literature on its efficacy).

Reviewer 4 Report

This manuscript addresses the long-term use of GnRH agonist implants, which now play a significant role in reproductive medicine in small animals. While the therapeutic possibilities resulting from the use of these implants are practically common knowledge, relatively little information is yet available on long-term effects and side effects. Therefore, this paper is of considerable interest to small animal veterinarians. There is little to criticize in terms of content. Streamlining of the manuscript is recommended to improve readability.

General comments

1) Occasional detail-rich presentations of normal findings, diagnostic measures, and treatments make the manuscript rather lengthy in places. The authors should check which of these details are really essential to support their statements.

2) The authors fall into clinician slang in places. Even if one understands what is meant, the authors should make an effort to use correct terminology (examples: "antiprostatic drug"; "biochemistry"; "chemiluminiscence" instead of "chemiluminiscence immunoassay"). Where appropriate, an abbreviation/acronym could be introduced.

Specific comments:

Line 7: "new molecule": see timing in heading and lines 58-61.

Lines 111-112: In this case, why not combine deslorelin application in the initial phase with an appropriate antibiotic, NSAID and osaterone acetate/cyproterone acetate (see e.g.: PMID: 33276394).

Lines 113-114: Finasteride should not be called an antiandrogen; it is a competitive inhibitor of steroid 5alpha-reductase. The importance of this enzyme in the pathogenesis of BPH should be mentioned somewhere in this manuscript. Finasteride does have a steroid-like structure, even though one carbon atom has been replaced by a nitrogen atom in the A ring of the steran backbone.

Line 204-205: missing here is an explanation of the molecular mechanism that counteracts UI.

Line 275: what is meant by "cortisol metabolism" here?

Line 286-287: to avoid confusion of the effect of a deslorelin implant with the negative feedback of sex steroids, "negative feedback" should be replaced here by "down-regulation of pituitary GnRH receptors".

Line 380: regarding aggressiveness, a clear distinction should be made here between aggressiveness towards other males in the context of sexual competition and aggressiveness towards puppies, bitches and humans.

Case 5: The statement "The two flare-up reactions occurring in 2019 and 2020 were due to the implant being used too close to its expiration date" as well as the recommendation "In order to avoid flare-up reactions in cases like this dog, it is important to schedule an appointment 1 month prior to the end of the previous implant's action in order to allow the dog's gonads to remain constantly under the effect of deslorelin" seem quite speculative. A corresponding reference is not to be found in the product information. Can faulty storage be excluded? Perhaps the reduced duration of effect was also due to a (now resolved) production-related problem. Perhaps an inquiry with the manufacturing company could provide clarity here.

Lines 540-542: the reported testosterone concentration is within the normal range of the male dog (however, in male dogs the testosterone secretion is subject to very strong almost pulsatile fluctuations during the day). It is also possible that the cause is on the side of the androgen receptor and downstream signaling cascades.

Lines 627-628: it would be interesting to know the basal value before GnRH stimulation.

Lines 649-651: the interpretation of this testosterone concentration is at least questionable for two reasons:

1) without knowledge of the measurement method used, its lower detection limit, and confirmation that this measurement method has been validated for use in the cat, reasonable interpretation of this result is not possible at all (for the problem of erroneous steroid measurements, see PMID: 28962971). For a reliable and sensitive assay, this concentration would be clearly suprabasal! Do not trust in commercial labs using kits established for humans and which have not validated in the respective target species!

2) Even in intact male cats, the testosterone concentration may temporarily drop below the detection limit of the usual testosterone assays (see PMID: 2391779).

Lines 651-659: the pronounced increase in testosterone after GnRH stimulation points with a high degree of certainty to the presence of endocrine-active testicular tissue, since possible extratesticular sources of testosterone (adrenal) would hardly have reacted in this way. Overall, the observations are clearly in favor of cryptorchidism; it is possible that the testis present was hypoplastic.

Regarding Case 7, it could be added that fibroadenomatosis may also occur in male cats when long-term progestogen preparations are used for "hormonal castration".

Typing errors:

Line 347: change glicemia to glycemia

Line 288: change deslorein to deslorelin

Reviewer 5 Report

The introduction provide sufficient background and include all relevant references. The research design is appropriate. The methods are described adequately. The results are clearly presented.

The paper needs minor revision in my opinion:

Part 3 (lines 720-784) should be called "Discussion"

After this part, Conclusions should be added. In my opinion, this may be a short note summarizing the entire study, such as: "The results of the study indicate that long-term use of dyslorein may be effective in diseases such as prostate hypertrophy, reproductive behavior disorders or urinary incontinence."

Author Response

Response to Reviewer 5 Comments

Point 1: Part 3 (lines 720-784) should be called "Discussion"

Response 1: Changed as requested.

Point 2: After this part, Conclusions should be added. In my opinion, this may be a short note summarizing the entire study, such as: "The results of the study indicate that long-term use of dyslorein may be effective in diseases such as prostate hypertrophy, reproductive behavior disorders or urinary incontinence."

Response 2: Thank you for your guidance, a fifth part has been added and briefly concludes the article after part 4, which is now titled “Discussion”.

Round 2

Reviewer 2 Report

Dear Authors, 

Thank you for the changes. As already mentioned, please change the non-English words in the figures. 

Author Response

Thank you for your observation, the ultrasound imagery has been updated as suggested.